# Spatial Trade-Offs and Temporal Evolution of Multiple Ecosystem Services in a Marine-Terrestrial Urban-Agglomeration Zone

**DOI:** 10.3390/ijerph17041231

**Published:** 2020-02-14

**Authors:** Huaxiang Chen, Lina Tang, Quanyi Qiu, Baosheng Wang, Weixiang Hu

**Affiliations:** 1Key Laboratory of Urban Environment and Health, Institute of Urban Environment, Chinese Academy of Sciences, Xiamen 361021, China; hxchen@iue.ac.cn (H.C.); lntang@iue.ac.cn (L.T.); bswang@iue.ac.cn (B.W.); 2University of Chinese Academy of Sciences, Beijing 100049, China; 3School of Marine Engineering, Jimei University, Xiamen 361021, China; wxhu@jmu.edu.cn

**Keywords:** spatial trade-offs, multiple ecosystem services, marine-terrestrial interlaced zone

## Abstract

It takes some time for changes to come in ecosystem services, and trade-offs occur in the process of changes. As opposed to a point in time, we use data spanning the years 2000–2005, 2005–2010, and 2010–2015 to study this research. After quantifying types of ecosystem services, this paper uses spatial correlation analysis and root-mean square deviation (RMSD) method to explore the relationships among ecosystem services and calculate the degree of spatial trade-offs (DT). Results show that the construction land increased substantially albeit at a declining rate of growth, and the degree of trade-offs (DT) increased with nontrivial differences in space. The hotspots for trade-offs are spatially aggregated in some areas but have varying patterns between ecosystem service pairs. The increasing area (IA) of construction land does not promote increased DT until the former reaches a certain threshold. With the exception of land use changes, type of industrial development is one of the key factors that influence the trade-offs of ecosystem services in the research region. We apply the models and methods used in this research to practice and discuss the practical value of our results for planners and decision makers vis-à-vis the design and instigation of appropriate development strategies.

## 1. Introduction

Marine-terrestrial interlaced zones are the heterogeneous zones between marine and terrestrial environments [1]. These zones are ecological ecotones known for their poor anti-interference capacity, fragile environment, low tolerance, and facile degradation compared to other types of land uses [2]. As coastal construction is in full swing, land occupation, environmental pollution, and other human factors are increasing, with one obvious factor being growing human populations and a concomitant huge increase in constructed land. These increasing associated impacts have a substantial impact on ecosystem services in terms of their structure and function [3], also influencing the quality and value of ecosystem services, creating the decline of ecosystem services. The Golden Triangle of Southern Fujian was a typical marine-terrestrial interlaced zone urban agglomeration with rapid urbanization in recent years. Significant amounts of agricultural land and forest areas have been lost because of urbanization, with varying impacts on ecosystem services per se [4].

Ecosystem services are environmental conditions and effects formed and maintained by the terrestrial and marine environment and the ecological processes therein; human beings depend on these services for wellbeing and survival [5,6]. Ecosystem services are diverse, with varying status such as trade-offs or synergies [7,8]. The trade-offs’ status may refer to the ever-evolving competition status. Trade-offs will occur when an ecosystem service is captured by one stakeholder at the expense of another, as multiple ecosystem services are competitively used [9]. There are four basic categories of trade-offs: spatial trade-offs, temporal trade-offs, trade-offs between beneficiaries, and trade-offs among ecosystem services [10]. The degree of trade-offs (DT) among ecosystem services equates to the strength of the ever-evolving competition. The higher the DT, the stronger the competition.

Managers and planners often make programming or policy decisions on a regular basis without the explicit consideration of these ecosystem service trade-offs or synergies [11]. It is particularly critical to explicitly analyze the potential trade-offs under different management scenarios since this will allow managers to minimize or eliminate losses to alternative services and formulate more effective, efficient, and defensible decisions [12].

Ecosystem service trade-offs have become a focal point of research in ecological economics in recent years [13]. Most of the relevant research can be divided into the following four categories: (1) Regional ecosystem service trade-offs and synergies [14,15,16]; (2) Impacts of land use changes on multiple ecosystem services [17,18,19]; (3) Land use optimization and management based on ecosystem service tradeoffs/synergies [20,21]; (4) Changes in individual ecosystem services under different land use scenarios [22]. Most of the research discussed above was based on a point in time during study periods, without considering that changes in ecosystem services take time to happen—trade-offs occur during the process of change. A lot of research used correlation analysis only to research the trade-offs, and some focused only on monetary values without thinking about the spatial biophysical units. These are precisely the issues covered in our study.

Appropriately identifying the ecosystem service trade-off types and features in a spatial-temporal context could help decision makers, not least urban planners, formulate rational strategies. Furthermore, the evolution of research in this domain means that guidelines for ecosystem assessments, conservation, and sustainable development are constantly being tested, validated, and improved where necessary [23,24].

This research aims to answer the following questions: (1) What changes have occurred in multiple ecosystem services’ spatial trade-offs in the Golden Triangle of Southern Fujian during the study periods? (2) What are the impacts to multiple ecosystem services’ spatial trade-offs and how can the acceleration be reduced?

## 2. Materials and Methods 

### 2.1. Case Study and Data Sources

The Golden Triangle of Southern Fujian lies on the southeastern coast of Fujian, China. It is a typical marine-terrestrial interlaced zone urban agglomeration with the fourth largest coastal economy in China [25,26]. It consists of three municipalities (Xiamen, Zhangzhou, and Quanzhou) and 28 county-level cities, counties, or districts (Figure 1; Appendix A, Figure A1). In 2015, the population of urban agglomeration was about 17 million and its Gross Domestic Product (GDP) accounted for 47.86% of the whole province [27]. Therein, the Quanzhou city is one of the starting points of China’s 21st Century Maritime Silk Road [25].

Temporally, it takes some time to have changes in ecosystem services, thus we operationalized the 2000–2015 data period in 5-year time-steps to start research. Spatially, 28 county-level cities, counties, or districts are considered. The data categories and sources used are listed in Table 1.

### 2.2. Methods

#### 2.2.1. Land Use

We use a model capturing the dynamic degree of land use to measure the intensity of land use change over time as well as differences across space [28,29]. The higher the absolute value of the dynamic degree, the more substantive are the land use changes. Positive (negative) values indicate growth (declines) in land use types (Equation (1)).
(1)$$K=\frac{U_{b}-U_{a}}{U_{a}\times T}\times 100\%$$

In Equation (1), K is the dynamic degree of one land use type in time period T; $U_{b}$ and $U_{a}$ are the final and initial areas of land use and land cover in study period, respectively; T is time in years.

The degree of land use (Equation (2)) refers to the scale and scope of human use of land for economic value through activities such as development, consolidation, and planning [28,29].
(2)$$L=100\times \overset{n}{\underset{i=1}{\sum }}\left(A_{i}\times C_{i}\right)$$

In Equation (2), L is the degree of land use (Dimensionless unit); $A_{i}$ is a classification index referencing each type of land use; $C_{i}$ is the proportion of land use to total land area referred for each type of land use; n is a classification number between 1 and 4 based on Wang [28], see Table 2.

Values are assigned according to the actual use of various types of land used by researchers.

#### 2.2.2. Ecosystem Services Assessment

Research into relationships between multiple ecosystem services are better identified and assessed by integrated social/ecological approaches than by either social or ecological data alone [30]. Overly based on the criteria of diversity, data accessibility, and representativeness, we choose eight indicators to do the research which covering supporting, provisioning, regulating, and cultural ecosystems services: habitat quality (HQ), net primary productivity (NPP), water provision (WP), food supply (FS), climate regulation (Carbon Storage, CS), water retention (WR), recreation services (RS), and landscape aesthetics (LA).

The reason for choosing these indicators is that the habitat quality (HQ) can quantify the extent to which ecosystems provide conditions species need to thrive. The net primary productivity (NPP) refers to the total amount of organic dry matter that green plants can produce per unit area [31]; it is an important indicator for evaluating the environmental quality of ecosystems [32]. The water provision (WP) refers to the potential capacity of surface water provisioning in a certain area. The more water that is provided, the better the potential water provision. The food supply indicator (FS) is a key societal task to safeguard effective food supply and promote nutritional balance to improve people’s health. The green plants play a critical role in terms of carbon sequestration which in turn is central to climate regulation (CS). The higher the carbon sequestration capacity, the stronger the climate regulation ability. The water retention (WR) refers to the process and ability of ecosystems to maintain moisture within a certain period and spatial area. The ecosystems could provide people with cultural-services, such as recreation (RS), relaxation, and experience of nature [33]. These services are important in urban ecosystems with significant economic value because they can improve human welfare in cities. Meanwhile, some ecosystems could produce landscape aesthetic benefits and this could potentially be valuable in landscape aesthetics (LA). The calculating methods are as follows (Table 3).

#### 2.2.3. Ecosystem Services Trade-offs

Using ArcGIS we standardize all spatial data and calculate spatial correlation coefficients in order to know the positive or negative relationship between pairs of ecosystem services. A negative relationship means a trade-off relationship exists between pairs of ecosystem services. Then, we use the indicator root-mean-square deviation (RMSD) method to quantify the trade-offs degree (DT) between ecosystem services [38]. As the value of RMSD increases, so too does the DT. The RMSD value can quantify spatial trade-offs between multiple ecosystem services; it extends the operationalization of trade-offs from negative correlations between ecosystem services to inhomogeneity rates of change in the same direction [14] and quantifies the average difference between standard deviations of individual ecosystem services and the average standard deviation of all ecosystem services [39], which describes the magnitude of scattering from the mean [14].
(3)$$DT=RMSD=\sqrt{\frac{\sum_{i=1}^{n}{\left(ES_{i}-\bar{ES}\right)}^{2}}{n-1}}$$

In Equation (3), DT is the degree of trade-offs between ecosystem services, n is the number of observations, $ES_{i}$ is the standardized value of the *i*th ecosystem services determined using fuzzy membership functionality in spatial analyst tools of ArcGIS to standardize the data. $\bar{ES}$ is the mean of ecosystem service standard values. In a two-dimensional coordinate system (Figure 2), $\bar{ES}$ is on the 1:1 line, when quantifying the trade-offs between two ecosystem services, the physical meaning of the RMSD value represents the distance from the coordinates of an ecosystem service pair to the 1:1 line or from the goodness of fit to the 1:1 line [14]. The relative position of a data point to the line indicates which ecosystem service is more beneficial [14]. The value of RMSD was divided into five levels according to natural discontinuity method (Table 4).

From the synergetic’s perspective, the coupling degree could determine the order and structure of the system when it reaches a critical region or the tendency of the system to go from disorder to order [40,41]. The key of the system from disorder to order lies in the synergy between the order parameters in the system, and this synergy influences the characteristics and laws of the phase transition. The coupling degree is just a measure of this synergy [42]. We used the coupling method to reflect the strength of interaction between increasing area (IA) of construction land and the DT between ecosystem services. The greater the extent of coupling, the stronger the interaction between the two [43].
(4)UA(ui)=(∏λ=1nui)1n
(5)C={[UA(u1)×UA(u2)]/[(UA(u1)+UA(u2))×(UA(u1)+UA(u2))]}12

In Equations (4) and (5), C is the coupling degree, n is the number of subsystems, $U_{A}\left(u_{i}\right)$ indicates general parameters of subsystems in the overall system, $U_{A}\left(u_{1}\right)$ is a comprehensive sequence of parameters pertaining to the increasing area (IA) of construction land, and $U_{A}\left(u_{2}\right)$ is the order parameter of ecosystem service.

We used the median segmentation method to demarcate the coupling degree into four intervals [41,44,45]. Specifically, 0 < C ≤ 0.3 corresponds to “low-level coupling”, 0.3 < C ≤ 0.5 is “mid-level coupling”, 0.5 < C ≤ 0.8 is “mid-level to high-level coupling”, and 0.8 < C ≤ 1 is “high-level coupling”.

## 3. Results

### 3.1. Impact of Urbanization on Regional Land Uses

#### 3.1.1. Contemporary Urbanization Characteristics and Trends

Over the research periods, the area of construction land increased significantly, and this increase was mainly concentrated in eastern coastal areas. The overall pattern of development also gradually increases from western to eastern coastal areas (Figure 3). The urbanization rate is greatest in Xiamen followed by Quanzhou and Zhangzhou. However, although urbanization has tended to proceed slower in Zhangzhou in most counties, it increased rapidly. The urbanization rates in Huli, Siming district in Xiamen, and Longwen, Xiangcheng district in Zhangzhou were close to 90% in 2015. The urbanization rates in Quangang, Fengze and Licheng, Luojiang district in Quanzhou were also high, exceeding 80%. With the exception of the Longwen and Xiangcheng districts of Zhangzhou, the other districts along the coast had high urbanization rates.

#### 3.1.2. Land Use Changes Influenced by Urbanization

The urbanization of Golden Triangle of Southern Fujian expanded rapidly, encroaching on and changing other types of land use and land cover. Through the matrix of land use transformation over 2000–2015, it is known that this urbanization mainly occurred by displacing agricultural land, forested land, and grassland, i.e., this urbanization was not brownfield development (Table 5). Among these areas, the water areas and unused land also increased during the study period, which is attributed to the deepening of the government and public emphasis on environmental protection. Many areas that were originally converted from water areas or unused lands into farmland or construction land have gradually been recovered as water areas or unused lands over the last 10 years.

The extent of land use transformation differs between cities and this is related to the intensity of urbanization. The overall rate of urbanization in Zhangzhou was relatively low; the transformation of land uses was therefore also small. By contrast, Quanzhou and Xiamen are intermediate and high in this respect. Though the transformation of land uses in Zhangzhou was lower than in Quanzhou in absolute terms, the intensity of change in construction land was higher in the former (Table 6). This is because Zhangzhou was at an early stage of urban development at the start of the study periods: urbanization proceeded quickly in proportional terms and the degree of land use changes was high; therefore, the dynamic degree of construction land was higher.

In 2000–2005, the dynamic degree of land use was highest in Zhangzhou, followed by Quanzhou, and then Xiamen. However, between 2005 and 2010 Xiamen scored highest in this respect, followed by Zhangzhou and then Quanzhou; in 2010–2015, the dynamic degree of construction land was highest in Quanzhou, followed by Zhangzhou and then Xiamen. The dynamic degree associated with construction land is highest followed by water areas, unused land, agricultural land, forested land, and grassland. During research time, the dynamic degree of construction land decreased, but the overall extent of land use degree increased, particularly in Xiamen followed by Zhangzhou and then Quanzhou (Figure 4).

### 3.2. Analysis of Spatial Trade-offs among Ecosystem Services

#### 3.2.1. Spatial Tradeoff Analysis

We calculated spatial correlation coefficients for each ecosystem service pair (Table 7). Furthermore, the results suggest that the following ecosystem service pairs exhibit a negative relationship and seem to have higher correlation coefficient values than other pairs of ecosystem services: recreation services-food supply (RS-FS), water provision-habitat quality (WP-HQ), water provision-carbon storage (WP-CS), water provision-net primary productivity (WP-NPP), and recreation services-net primary productivity (RS-NPP). Only the habitat quality-carbon storage (HQ-CS) pair exhibits a positive relationship.

The tradeoff relationships and DT among ecosystem service pairs varied over time and space. In all the ecosystem services, most areas in the Golden Triangle of Southern Fujian had a weak DT in 2000–2005, and some changed to low moderate degree in 2005–2010, but in 2010–2015, most areas in the east were changed to low moderate DT (Figure 5). In pairs of NPP-WP and NPP-RS, there were more areas that had a moderate or strong DT in 2010–2015.

Overall, the DT values of ecosystem services in 2010–2015 were higher for Zhangzhou, followed by Quanzhou and Xiamen. The spatial distributions of DT between water provision-habitat quality (WP-HQ) and water provision-carbon storage (WP-CS) were similar, as the DT in 2005–2010 was higher than that in 2010–2015 and 2000–2005. Further, the DT in Zhangzhou was generally higher than that in Xiamen and Quanzhou. In addition, the DT between water provision-net primary productivity (WP-NPP) increased over the study periods, particularly in southeast coastal counties. For the food supply-recreation service (FS-RS) in 2000–2005, the DT in Zhangzhou was substantially lower than those found in Xiamen and Quanzhou. For the recreation service-net primary productivity (RS-NPP), in 2000–2005, the DT in Quanzhou were generally lower than in Zhangzhou and Xiamen.

#### 3.2.2. Hotspot Analysis

To determine the key areas of higher DT between ecosystem service pairs, we apply a hotspot analysis through the ArcGIS platform Spatial Statistics Tools-Hot Spot Analysis (Getis-Ord Gi*). Furthermore, we find that the hotspots with similar confidence levels spatially aggregate in some areas, but the patterns between ecosystem service pairs remain varied (Figure 6). For the selected ecosystem services, the hotspots mainly form in Nan’an city of Quanzhou in 2000–2005, but migrate to the central area of Zhangzhou city in 2010–2015. Only the pairs CS-WP and HQ-WP were similar to each other; the hotpots were spatially scattered in the west in 2000–2005, then gradually moved to the center of the urban agglomeration in 2005–2010, until eventually permeating the entire mid-west of Zhangzhou City in 2010–2015. For the pair RS-FS, the hotspots mainly formed in Nan’an county in 2000–2005, in Nan’an, Pinghe county in 2005–2010, and in Pinghe and Anxi counties in 2010–2015. The hotspots for RS-NPP were mainly distributed in coastal areas during the research periods.

### 3.3. Impacts of Changes in Increasing Areas of Construction Land on DT among Ecosystem Services

To understand whether and how the increasing areas (IA) of construction land was related to the DT between ecosystem services, we conducted analysis using a coupling method [41].

The coupling degree between IA of construction land and DT in ecosystem services slightly decreased over the study periods but does not go lower than “mid-level coupling” and tends to “mid-level to high-level coupling” (Table 8). This suggests that the DT in ecosystem services was related to the increasing area (IA) of construction land.

To prevent effects from small samples to coupling results and explore differences in DT among ecosystem services between counties with higher (above 80%) and lower (below 80%) urbanization rates under different urbanization scenarios, we constructed scatter plots between DT and IA through tool of Statistical Product and Service Solutions (SPSS, https://www.ibm.com/products/spss-statistics) (IBM company, Chicago, United States). The plots (Appendix A, Figure A2) suggest a certain correlation between the two variables and as the increasing area of construction land varies across counties, so too does its impacts on DT among ecosystem services.

## 4. Discussion

### 4.1. Why such Trade-offs Occur between Ecosystem Services

Based on our previous research, four types of industrial development exist in the Golden Triangle of Southern Fujian: agricultural-led, industrial-led, business services, and balanced development [41]. Combined with the results of the present work, we find that counties with a high DT among ecosystem services mainly experience agricultural-led and industrial-led development, whereas counties with balanced development have a lower DT among ecosystem services. In particular, the pairs of net primary productivity-water provision (NPP-WP) and recreation service-net primary productivity (RS-NPP), which have a higher DT among ecosystem services, mainly correlate with Quanzhou’s industrial-led counties and Zhangzhou’s agricultural-led counties.

Why do such trade-offs or synergy relationships occur? More forest land results in higher habitat quality and net primary productivity but lowers the amount of surface water, or it results in higher potential value of recreation services but lowers net primary productivity. The land use changes will have impacts on ecosystem service material amounts, influencing the values of ecosystem services. Some types of ecosystem services increase in material amounts and some decrease (as do the values); therefore, the relationship between different types of ecosystem services changed, resulting in trade-offs.

With the process of urbanization and development of industry in the research region, the agricultural land, forest land, and grassland decrease and the construction land and water area increase, having an impact on the change of ecosystem services and further influencing the trade-offs or the synergy relationship among ecosystem services. For instance, the increases of construction land in east coast urban areas promote the provision of surface water but lower the quality of habitat and the amount of carbon storage, making a tradeoff between water provision and habitat quality (WP-HQ) and water provision and carbon storage (WP-CS).

### 4.2. Impact on Spatial Ecosystem Service Trade-offs

The increase in construction land is an important factor of the urbanization process. In our research, the increasing area (IA) of construction land in different counties impacts the DT among ecosystem services, but the degree of impact is varied and the effect of trade-offs for different types of ecosystem services also differs. The present results mainly reveal the influence of trade-offs between the pairs WP-NPP and RS-NPP. The DT between ecosystem services in 2010–2015 was clearly larger than that in 2000–2005, and the changes mainly occurred in coastal areas, which is attributed to human demand for recreation services and a willingness to pay for leisure and entertainment that greatly increased due to the increased income in coastal areas (the Gross Domestic Product per capita increased by 66% in 2010–2015, which far exceeds the 23% in 2000–2005). Thus, the potential value of ecosystem recreation service increased. However, the net primary productivity could not keep up with demand, leading to higher trade-offs in 2010–2015 within coastal areas.

The impact of increasing areas (IA) for construction land to ecosystem service trade-offs was different. For example, we compared the results of this study with Wang’s [46] cognate research, which found that the tradeoff relationship from Wang was mainly concentrated in the food supply-hydrological regulation service pair during 2005–2010 in urban-rural fringe areas; in 2010-2015, the trade-off or synergy relationships were mainly concentrated among food supply, hydrological regulation, soil conservation, and maintenance of nutrient cycling. In our study, spatial synergies were most apparent between the habitat quality-carbon storage (HQ-CS) whilst spatial trade-offs were revealed with respect to water provision-habitat quality (WP-HQ), water provision-carbon storage (WP-CS), recreation service-food supply (RS-FS), and recreation service-net primary productivity (RS-NPP).

The reason for this difference is the choice of spatial scale and difference of the calculation methods. Our RMSD method refers to the average difference between standard deviations of individual ecosystem services and the average standard deviation of all ecosystem services [39]. It extends the operationalization of trade-offs from Wang’s negative correlation relationship (from correlation analysis) to inhomogeneity rates of change in the same direction [39]. From another perspective, our study tends to indicate the trade-offs in space among multiple ecosystem services based on the analysis of correlation.

### 4.3. How to Reduce Acceleration in Ecosystem Service Trade-offs

The DT of RS-NPP and WP-NPP initially decrease with the increasing areas (IA) of construction land across all counties, then increase when the IA reaches a threshold. However, other ecosystem service pairs increase with the IA of construction land, then decrease with continuing IA. The DT then increases again when the increasing area reaches a threshold (Appendix A, Figure A2). It can be inferred that changes in IA of construction land impact the DT among ecosystem services. Generally, increasing IA of construction land does not contribute to increasing DT in early urban development, but from the mid period to the late period, the opposite is the case. Moreover, the threshold in counties with urbanization rates higher than 80% (i.e., the mid-to-late period of urban development) was far lower than that in counties with urbanization rates lower than 80% (early to mid-term period of urban development). The reason for this pattern is that counties with medium or low levels of urbanization have been less disturbed by humans and increasing IA does not negatively affect DT until higher levels of urbanization are reached.

Since the increasing area (IA) of construction land promoted the DT among ecosystem services during urbanization in the mid-to-late stages, how to reduce the acceleration to ecosystem service trade-offs in research areas was a matter of concern. Our research suggests that one method is to grasp the construction land and ecosystem services scope when spatial trade-offs appear. In other words, we pay attention to the ecosystem service pairs NPP-WP, RS-NPP, and WP-HQ, WP-CS to ensure sustainability between urbanization and ecosystem services. For example, in areas with high DT between net primary productivity-water provision (NPP-WP), we can promote innovation in agricultural production technologies on limited agricultural land. Furthermore, in areas with high DT between recreation service-net primary productivity (RS-NPP), we can develop the leisure and the entertainment industry combined with local folk culture with a view to increase the economic and ecological value per unit area and reduce the occupancy and impact of recreation service on NPP.

During study periods, the DT between ecosystem services increased over time and varied spatially across counties. Urban planning should be sensitive to multi-dimensional demands from different stakeholder groups and be suitably constrained by existing policies. Within the Golden Triangle of Southern Fujian, Xiamen can be defined as business-services oriented, Zhangzhou is agriculturally oriented, and Quanzhou is industrially oriented [41]. So, it is necessary to keep the amount of agricultural land in Zhangzhou and develop a model of eco-agriculture. In addition, it is necessary to increase the population of labor and personal economic income to decrease the DT between recreation service-net primary productivity (RS-NPP) and recreation service-food supply (RS-FS). However, in Xiamen city, we can reduce the DT between water provision-habitat quality (WP-HQ) and water provision-carbon storage (WP-CS) based on increased green area under the urbanization process. The same methods can be used in Quanzhou city to reduce the acceleration of ecosystem service trade-offs.

### 4.4. Limitations

In summary, this paper analyzed spatial ecosystem service trade-offs in the Golden Triangle of Southern Fujian, calculated the DT among ecosystem services, found the hotspots of spatial trade-offs, and analyzed the relationship between increasing area (IA) and DT in ecosystem service pairs. This led to practical advice for urban planners and other decision makers. However, in common with applied research more generally, this research is not without its limitations and addressing these limitations could be fruitful terrain for future research. First, there is ample scope to consider a broader indicator set, to incorporate uncertainty analysis into the quantitative methodology employed, and to augment the research design by instigating a mixed-methods approach which uses qualitative data and techniques to tackle the issues and questions explored herein. Second, the data samples were too few to support and verify the threshold at which IA in construction land begins to affect the DT among ecosystem services. More data and cases are needed to verify the accuracy and rationality of the specific thresholds.

## 5. Conclusions

There are significant differences in levels of development between counties in the Golden Triangle of Southern Fujian, which lies in the marine-terrestrial interlaced zone. Trade-off relationships among ecosystem services are also spatially heterogeneous. Through this research, we have some findings.

(1) Although the overall ecosystem service trade-offs in the Golden Triangle of Southern Fujian were in the low-to-moderate range, they increased over the years of the study and continue to increase.

(2) The trade-offs occur not only between supplying services and supporting services but also between supplying services and regulation services, and cultural services and supporting services.

(3) The DT of ecosystem service pairs RS-NPP (recreation services-net primary productivity) and WP-NPP (water provision-net primary productivity) increased more in 2010–2015 than in 2000–2005. This should alert urban planners and other decision makers to pay attention to trade-offs among these ecosystem services.

(4) Urbanization affects ecosystem service spatial trade-offs, with the increasing area of construction land and the types of industrial development being two of the key factors.

(5) There is a nonlinear relationship between changes in increasing area (IA) of construction land and DT among ecosystem services, with the former impacting the latter only after a threshold value of IA was exceeded. Again, this is salient information for relevant stakeholders.

## Figures and Tables

**Figure 1 ijerph-17-01231-f001:**
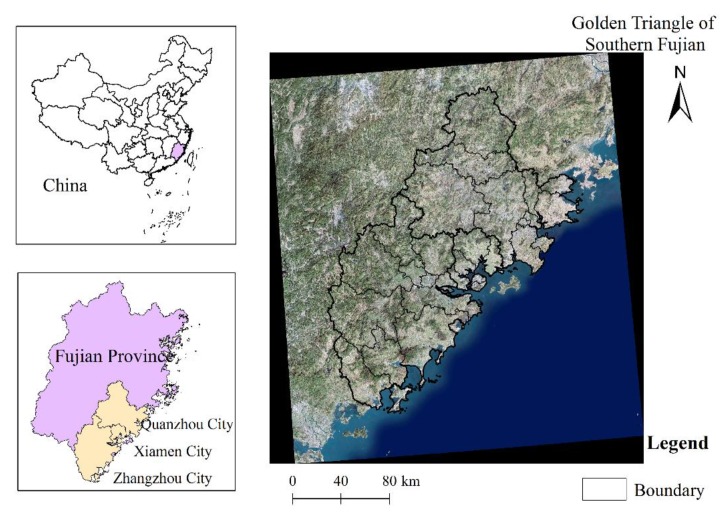
Geography of the Golden Triangle of Southern Fujian.

**Figure 2 ijerph-17-01231-f002:**
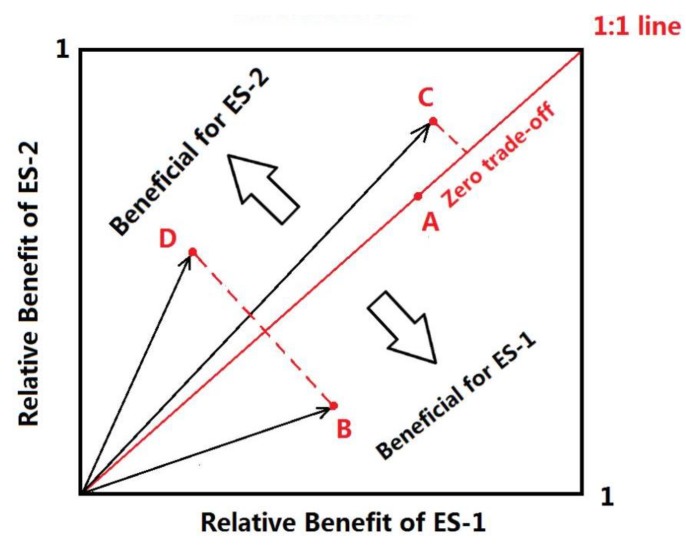
Description about the trade-offs between two ecosystem services [14].

**Figure 3 ijerph-17-01231-f003:**
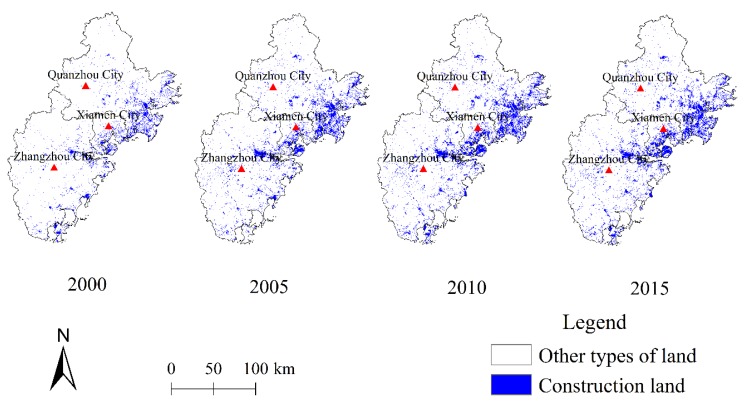
Spatio-temporal urbanization trends (Unit: grid).

**Figure 4 ijerph-17-01231-f004:**
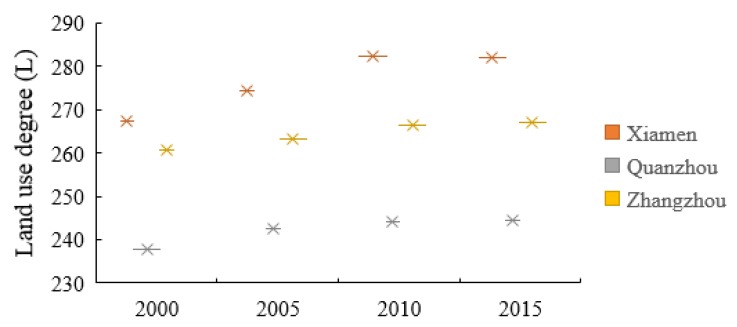
Index of overall land use degree between 2000 and 2015 (dimensionless).

**Figure 5 ijerph-17-01231-f005:**
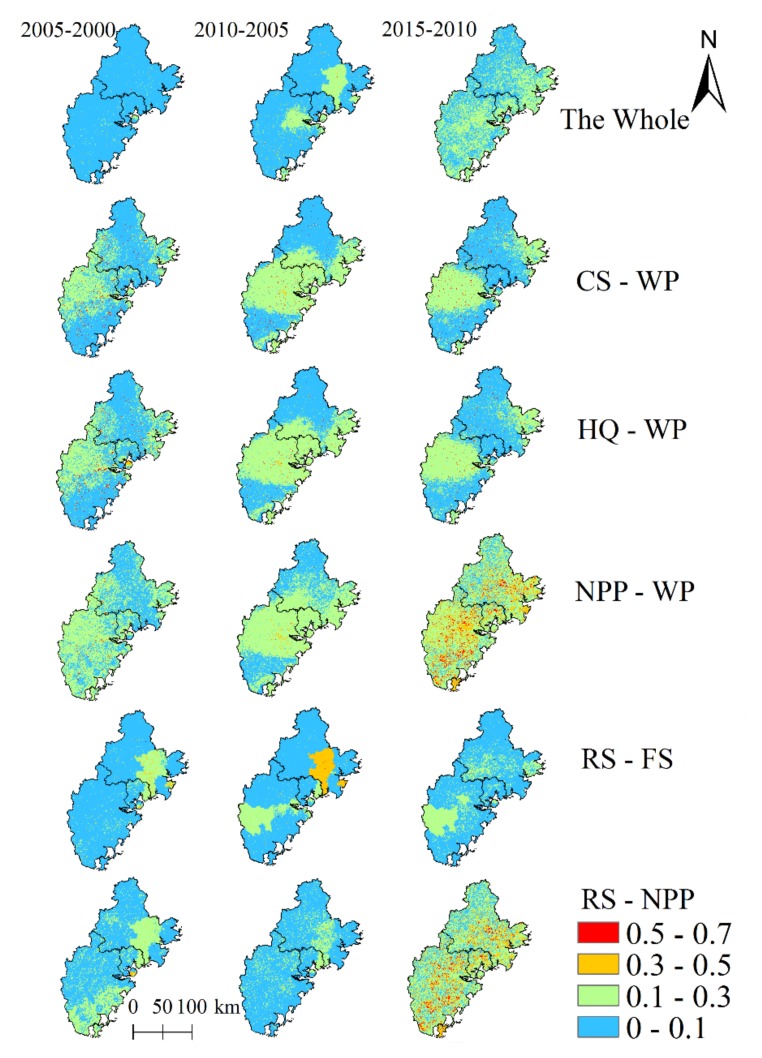
Spatio-temporal differences in trade-offs degrees between ecosystem service pairs (dimensionless).

**Figure 6 ijerph-17-01231-f006:**
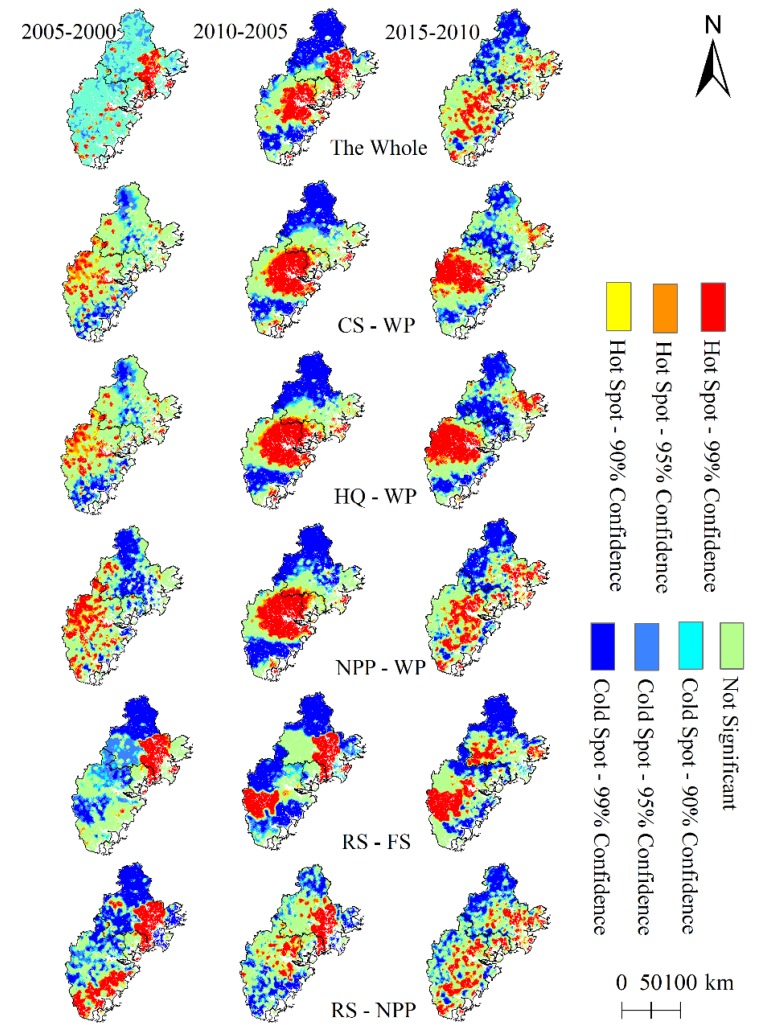
Hotspots of trade-offs degree with different confidence levels among ecosystem services.

**Table 1 ijerph-17-01231-t001:** Data categories and sources.

Data Category	Description	Sources
Statistical data	Data about the Gross Domestic Product (GDP), Population, Crop production.	Fujian Statistical Yearbook (2001, 2006, 2011, 2016)
Land use and land cover data NPP data	1-Cultivated land, 2-forestland, 3-grassland, 4-water area, 5-construction land, 6- unused land The net primary productivity data	Resource and Environment Data Cloud Platform, China (http://www.resdc.cn/Default.aspx)
Weather data	Data about the temperature, precipitation.	National Meteorological Information Center, China (http://data.cma.cn/)
Soil data	Data about the soil depth, soil sand content (%), soil silt content (%). soil clay content (%), organic carbon (%), and soil bulk density (g/cm^3^)	World Soil Database _ China soil Dataset
DEM data	Data about the slope, elevation.	Geospatial Data Cloud of China (http://www.gscloud.cn/)
Other data	Data about the soil saturated hydraulic conductivity (cm/d), flow coefficient V.	All from others’ research (shown in the Methods section)

**Table 2 ijerph-17-01231-t002:** The index of classification.

Types	Level of Unused Land	Level of Grassland, Forested Land, and Water Areas	Level of Agricultural Land	Level of Construction Land
Index of classification	1	2	3	4

Note: The index of classification is different than the code of land use classification since it is an index.

**Table 3 ijerph-17-01231-t003:** Calculating methods of indicators.

Indicator	Method	Description
Habitat quality (HQ)	$Q_{x}=H_{j}\left[1-\left(\frac{D_{x}^{z}}{D_{x}^{z}+k^{z}}\right)\right]$ From the Integrated Valuation of Ecosystem Services and Tradeoffs model, InVEST. (https://naturalcapitalproject.stanford.edu/software/invest)	$Q_{x}$ is habitat quality in grid x, $D_{x}^{z}$ is the level of threat in grid x, x is a half-saturation constant, which usually equates to half of the maximum value of $D_{x}^{z}$, $H_{j}$ is the habitat suitability of the jth land use and land cover, and z is a normalized constant, which is usually taken as 2.5.
Net primary productivity (NPP)	Resource and Environment Data Cloud Platform, China (http://www.resdc.cn/Default.aspx)	
Water provision (WP)	$Y_{x}=\left(1-AET_{xj}/P_{x}\right)P_{x}$ From the Integrated Valuation of Ecosystem Services and Tradeoffs model, InVEST. (https://naturalcapitalproject.stanford.edu/software/invest)	$Y_{x}$ is average annual water provision in grid x, $P_{x}$ is average annual precipitation in grid x, $AET_{xj}$ is average annual evaporation in grid x of the jth land use and land cover type.
Food supply (FS)	According to China’s Food and Nutrition Development (2014–2020) [34], annual per capita food consumption in China in 2020 will be 135 kg (12 kg edible vegetable oil, 13 kg beans, 29 kg meat, 16 kg eggs, 36 kg milk, 18 kg aquatic products, 140 kg vegetables, and 60 kg fruit)	We can determine the number of people that can be supplied with food in each county according to China’s Food and Nutrition Development (2014–2020).
Carbon storage (CS)	$C_{\_\mathrm{to}tal}=C_{\_above}+C_{\_below}+C_{\_soil}+C_{\_dead}$ From the Integrated Valuation of Ecosystem Services and Tradeoffs model, InVEST. (https://naturalcapitalproject.stanford.edu/software/invest)	$C_{\_\mathrm{to}tal}$ is total carbon storage, $C_{\_above}$ is carbon storage above ground, $C_{\_below}$ is carbon storage below ground, $C_{\_soil}$ is soil organic carbon stock, $C_{\_dead}$ is dead (litter) organic carbon stock.
Water retention (WR) (Quantitatively, the residence time of water is calculated per grid unit based on water production data, topographic index data, soil saturated hydraulic conductivity data, flow coefficient data, and so on)	$R=min\left(1,\frac{249}{V}\right)\times min\left(1,\frac{0.9\times TI}{3}\right)\times min\left(1,\frac{U}{300}\right)\times Y_{x\;}$ $TI=log\left(\frac{D}{S\times P}\right)$ $U=1.157\times 10^{-4}\times exp\left(x\right)$ $x=16.75-2.333\times log\left(sand\right)-1.303\times log\left(silt\right)-0.074\times clay-1.688\times log\left(som\right)+3.605\times som-11.106\times log\left(\rho_{b}\right)$ See Wang [35].	R is the amount of water retention (mm), V is a flow coefficient which can be derived from the literature [35], TI is the topographic index which can be calculated through DEM, U is soil saturated hydraulic conductivity (cm/d), and $Y_{x}$ is average annual water provision in grid x. D is the number of watershed cells in a watershed unit, S is soil depth, and P is the percentage of slope. sand denotes soil sand content (%), silt is soil silt content (%). clay is soil clay content (%), som is the distribution of soil organic matter which can be replaced by organic carbon (%), and $\rho_{b}$ is soil bulk density (g/cm^3^).
Recreation services values (RS)	Based on the results of Cao [33], we use a correction coefficient to estimate the willingness to pay for recreation services in the study area.	
Landscape aesthetic values (LA)	Based on the results of Costanza [36] and Xie [37], we apply a correction coefficient to estimate the aesthetic value of ecosystem services in the study area.	

**Table 4 ijerph-17-01231-t004:** Classification of synergies and trade-offs between ecosystem services.

	Degree of Synergy	Range	Degree of Tradeoff
[0, 0.1]	Weak	[−0.1, 0]	Weak
[0.1, 0.3]	Low Moderate	[−0.3, −0.1]	Low Moderate
[0.3, 0.5]	Moderate	[−0.5, −0.3]	Moderate
[0.5, 0.7]	Strong	[−0.7, −0.5]	Strong
[0.7, 0.9]	Very Strong	[−0.9, −0.7]	Very Strong

**Table 5 ijerph-17-01231-t005:** Shifting land use in the Golden Triangle of Southern Fujian between 2000 and 2015 (Unit: ha).

	2000	Agricultural Land	Forested Land	Grassland	Water area	Construction Land	Unused Land	Total
2015								
Agricultural land		537,354	10,425	3405	576	2710	19	554,489
Forested land		10,443	1,224,706	12,555	397	986	47	1,249,134
Grassland		3985	13,768	376,367	129	420	149	394,818
Water area		4470	1762	552	35,163	8964	26	50,937
Construction land		80,363	33,567	14,439	4336	114,420	55	247,180
Unused land		27	82	169	109	3	1964	2354
Total		636,642	1,284,310	407,487	40,710	127,503	2260	2,498,912

Note: It can be read as 33,567 ha of forest land in 2000 were converted to construction land in 2015.

**Table 6 ijerph-17-01231-t006:** Dynamic degree of land use between 2000 and 2015.

	2015–2000				2015–2010				2010–2005				2005–2000			
	Whole Region	Xia-men	Quan-zhou	Zhang-zhou	Whole Region	Xia-men	Quan-zhou	Zhang-zhou	Whole Region	Xia-men	Quan-zhou	Zhang-zhou	Whole Region	Xia-men	Quan-zhou	Zhang-zhou
Agricultural land	−0.86	1.60	−0.93	−0.63	−0.22	0.04	0.36	0.11	0.88	1.97	0.71	0.82	1.58	3.12	1.81	1.00
Forested land	0.18	0.17	−0.15	0.21	0.00	0.02	0.01	0.00	0.15	0.15	0.13	0.16	0.41	0.39	0.32	0.48
Grasslands	−0.21	0.20	−0.21	−0.20	0.00	0.04	0.00	0.01	−0.08	0.11	0.00	−0.13	−0.55	0.53	−0.63	−0.49
Water areas	2.11	1.30	1.29	4.14	1.39	0.25	1.64	1.29	4.13	1.97	1.56	7.82	0.41	3.18	0.45	1.88
Construction land	6.27	6.44	5.76	6.96	0.45	0.13	0.63	0.35	4.32	6.98	2.89	5.33	11.21	8.98	11.56	11.71
Unused land	1.15		0.75	35.28	−0.02		−0.02	−0.02	3.09		2.72	147.31	0.32		−0.38	−4.95

**Table 7 ijerph-17-01231-t007:** Correlation coefficients for pairs of ecosystem services.

	WP	WR	RS	NPP	LA	HQ	FS	CS
WP	1.00	−0.01	0.12	−0.22	0.01	−0.52	−0.05	−0.43
WR		1.00	0.02	−0.02	0.03	0.02	0.01	0.02
RS			1.00	−0.19	0.03	−0.02	−0.54	−0.01
NPP				1.00	−0.06	0.03	0.06	0.02
LA					1.00	0.08	−0.05	0.05
HQ						1.00	0.00	0.80
FS							1.00	0.00
CS								1.00

**Table 8 ijerph-17-01231-t008:** Coupling degree between increasing areas (IA) of construction land and degree of trade-offs (DT) of ecosystem services.

	2000–2005	2005–2010	2010–2015
All Ecosystem Services	0.49	0.49	0.48
RS-FS	0.50	0.50	0.50
RS-NPP	0.50	0.49	0.48
WP-CS	0.50	0.50	0.49
WP-NPP	0.50	0.50	0.45
WP-HQ	0.49	0.49	0.49

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
