# Peer review of "Spatial Trade-Offs and Temporal Evolution of Multiple Ecosystem Services in a Marine-Terrestrial Urban-Agglomeration Zone"

_ijerph, 2020, doi:10.3390/ijerph17041231_

Round 1
Reviewer 1 Report
This paper did good research on the ecosystem services and trade-offs in the marine-terrestrial interlaced zones. Here are some suggestions:
the introduction should be improved, show more details of originality.
the trad-offs methods need more explanation, what do these indexes really show?
"C"in line 142-143 had different format with the former equation, are these "C" same?
the result had better show more details in the temporal variation of ES and trad-offs. Fig 5 needs improve
Author Response
Dear Reviewer,
Thank you for reviewing our manuscript. Your comments helped us to revise and improve the paper. We considered your comments carefully and made corrections accordingly. Please see below the cover letter for main responses to the comments.
Thank you again for reviewing our manuscript. We look forward to your feedback once more.
Best regards.

Reviewer 2 Report
See attached.

Author Response
Dear Reviewer,
Thank you for reviewing our manuscript. Your comments really helped us to revise and improve the paper. We considered your comments carefully and made corrections accordingly. In addition, we had the revisions edited by a professional science editor to improve the writing. Please see below the cover letter for responses to the comment.
Thank you again for reviewing our manuscript. We look forward to your feedback once more.
Best regards.
